# A Reinforcement Learning Control in Hot Stamping for Cycle Time Optimization

**DOI:** 10.3390/ma15144825

**Published:** 2022-07-11

**Authors:** Nuria Nievas, Adela Pagès-Bernaus, Francesc Bonada, Lluís Echeverria, Albert Abio, Danillo Lange, Jaume Pujante

**Affiliations:** 1Eurecat, Technology Centre of Catalonia, 08005 Barcelona, Spain; francesc.bonada@eurecat.org (F.B.); lluis.echeverria@eurecat.org (L.E.); albert.abio@eurecat.org (A.A.); danillo.lange@eurecat.org (D.L.); jaume.pujante@eurecat.org (J.P.); 2Business Administration Department, Universitat de Lleida, 25001 Lleida, Spain; adela.pages@udl.cat

**Keywords:** hot stamping, reinforcement learning, autonomous control

## Abstract

Hot stamping is a hot metal forming technology increasingly in demand that produces ultra-high strength parts with complex shapes. A major concern in these systems is how to shorten production times to improve production Key Performance Indicators. In this work, we present a Reinforcement Learning approach that can obtain an optimal behavior strategy for dynamically managing the cycle time in hot stamping to optimize manufacturing production while maintaining the quality of the final product. Results are compared with the business-as-usual cycle time control approach and the optimal solution obtained by the execution of a dynamic programming algorithm. Reinforcement Learning control outperforms the business-as-usual behavior by reducing the cycle time and the total batch time in non-stable temperature phases.

## 1. Introduction

Hot stamping of sheet steel, also known as press hardening, is a non-isothermal forming process for sheet metals that allows obtaining lightweight components with complex shapes and very high mechanical properties. In this hot sheet metal forming process, forming and quenching takes place in the same forming step [1]. Ultra-high strength steels are demanded in the construction of structural elements and safety-relevant components in the automotive industry in order to reduce vehicle weights and improve safety and crashworthiness qualities [2].

The process follows the following steps: (i) sheet steel is heated to austenitization temperature, typically in the 890–950 °C range, (ii) transferred to a refrigerated set of dies installed in a press, and (iii) forming (shape pressing to the heated sheet with low strength) and quenching (fast cooling to transform the sheet into strong and stable martensite) in a single step [3]. This combination of forming and final quench in a single operation is the defining point of hot stamping [4]. If we compare this technology with cold or warm stamping processes, lower force and lower power are required, more plastic deformation is allowed, parts and tool damage problems are reduced, and microstructure is improved in the optimal deformation temperature range due to recrystallization of deformed grains [5]. 

The final part quality and its high properties are affected by the different parameters set during the process and the forming material. The time spent on transfer, forming, and consecutive quenching phases represents a cycle. Process cycle parameters are usually set up based on experience, combined with limited trial-and-error at line set up with the support of finite element modeling software for offline (slow) experimentation. Cycle times are defined at tool setup and are kept constant during batch production. Therefore, one of the main concerns in hot stamping processes, which are continuously facing increasing demands, is the process time reduction to allow higher productivity, maintain quality, and minimize costs and raw material losses [2]. 

Optimal control is developed to find the most efficient ways to outperform human decision-making in dynamic and uncertain systems based on cost function definitions that represent the relevant concerns and desired final goals. Decision-making and control problems have been traditionally faced with human-based expertise, rule-based techniques, mathematical optimization, metaheuristic, and heuristics. For instance, genetic algorithms in metaheuristics have been widely employed for multi-objective process parameters optimization in hot stamping production, ensuring forming quality [6], considering stochastic variability of these parameters [7], and reducing energy consumption [8]. Classical optimal control solutions are considered offline, require complete and previous knowledge of the environment dynamics, and are not able to react to unexpected changes and handle uncertainties favorably. The lack of these methods for real-time, uncertain, and changing control constraints their applicability to more dynamic and open environments [9]. Thus, to better control complex and changing systems under uncertainties, more adaptive control is needed. The digital transformation, the increasing use of more complex information systems, and the constant research on new and more advanced technologies are producing changes in the manufacturing industry’s organization, development, and control. Machine Learning (ML) field within Artificial Intelligence is becoming extensively explored to find the most suitable applications in manufacturing, such as quality prediction, predictive maintenance, and process control, to support informed decision making. Industrial environments are uncertain and dynamic, with continuously changing resources, constraints, and demands. Therefore, manufacturing processes, with all these new and uncertain variables at stake, cannot be operated or controlled optimally; thus, adaptability represents an essential ability to guarantee the desired success. Data-driven models can extract knowledge, expert behavior, historical tendencies, and underlying information from collected system data and learn from it [10]. These methods can be used to maximize process Overall Equipment Efficiency, reduce operational costs, climate footprint, and any other performance indicator or valuable metric, and make a system autonomous from a human interaction perspective. 

The principal approaches under the ML domain are Supervised Learning, Unsupervised Learning, and Reinforcement Learning (RL) methodologies. While RL has been mainly used in robotics, scheduling, gaming, and navigation areas, the other types of ML approaches have all been widely proposed in many different and diverse industrial and manufacturing domains for optimization tasks, quality forecast, predictive maintenance, and soft sensors. In particular, T. Hart-Rawung et al. [11] applied an Unsupervised Learning model based on artificial neural networks (ANN) to calculate the hot-stamped parts final phase of hot stamping tools design. Supervised Learning has successfully been considered in hot metal forming systems for manufacturing feasibility prediction [12], anomaly detection [13], and maximum thinning and thickening rates prediction of hot-stamped parts [14]. RL represents a set of solutions that do not previously need to know any information about the system dynamics, in contrast with other traditional control and optimization techniques, and give immediate real-time answers to any faced situation. The adaptive, generalization, and immediacy nature of RL methods offer great potential to be used as a decision support system or to directly manage, in an autonomous manner, decision processes. However, RL learns from sequential data generated by trial-and-error, and its application is challenging in complex industrial control tasks, where the consequences of bad decisions can be very costly. Therefore, process simulators must be considered to overcome this challenge. RL methods require collecting a huge number of experiences through interactions with the systems to learn the complex dynamics from scratch [15]. This collection step is time-consuming. Despite RL capabilities, the difficulties presented are the main causes of the delayed application of RL to industrial manufacturing environments.

We highlight recent RL approaches developed for process planning and pass schedules design in heavy plate rolling processes [16,17] and open die forging processes [18]. Recently, Liu et al. [19] have applied an RL method to free-from sheet-metal stamping. The process transition data used for training have been provided through finite element simulations. Moreover, they identified the limitation of the simulation speed for more complex cases. Variability in cycle time and its optimization are not considered in this study.

In the present work, an RL-based system is trained to master the production of batches, producing high-quality components, with dynamic cycle time optimization being the ultimate goal. The training and evaluation are made within a simulation environment of surrogate models based on the simulation software ABAQUS [20]. To the best of the authors’ knowledge, there are no RL applications for dynamic optimization of the cycle time in batch production. The RL application learns by interacting with a simulator of the environment. The industrial process simulator is made by supervised learning models presented in more detail in [21]. A self-experienced interaction-driven system based on RL is compared with an optimal dynamic programming (DP) solution in a discretization of the process variables, showing promising results for the consideration of an extension of RL solutions to more complex systems where DP cannot be applied. Therefore, the main contribution of this paper is a novel application of RL in a hot stamping process without considering any previous knowledge and using surrogate models as the environment simulator. This production optimization method serves as a proof-of-concept which can be easily extended to more complex processes and more elaborated RL methods.

This paper is organized as follows. In Section 2, the hot stamping process is presented, a brief introduction to the DP and RL methodologies is given, and a detailed problem formulation is provided. The description of the process optimization experiments with DP and RL and their results are presented in Section 3. In Section 4, we compare the performance of the algorithms and discuss the results. Finally, conclusions and further work are displayed in Section 5.

## 2. Materials and Methods

In this section, the principal concepts on which the research is based are presented. Firstly, the existing studied industrial challenge in hot stamping processes is described in detail. Secondly, the dynamic programming and Reinforcement Learning methodologies compared are theoretically explained in the context of Markov Decision Processes (MDP). Finally, the specific problem formulation by relating the needs in the production process in the MDP scheme is given. 

### 2.1. Hot Stamping Process

The hot stamping process is a thermo-mechanical metal forming process used to generate complex, light, and resistant products in fewer steps than other traditional stamping methods. This process is based on the metallographic characteristics’ transformation phase of the metal sheet simultaneously with its shaping phase. Initially, the blank has low mechanical properties, and, due to the high temperatures in which the blank is heated: (i) very complex shapes can be achieved, and (ii) the mechanical properties are improved as a result of the recrystallization process, increasing its ductility and preventing the formation of microcracks.

There exist two performing techniques used in hot stamping processes for blank transformation into part: the direct method and the indirect method. This work is focused on the direct method (see Figure 1), consisting of heating the sheet at a very high temperature until the structure remains completely austenitic and transferring the hot blank to a die which is closed immediately for simultaneous forming and quenching phases. Finally, the part is obtained and removed from the die with the desired shape and mechanical properties. This method, which allows mass production, and thus incurs lower production costs, is the most used in industrial production. The real-time optimization approach presented in this paper is based on this method. A comprehensive summary of this technique can be found in reference [2].

Since the quality and high properties in the transformed part must be ensured, and there is variability in some process conditions such as temperatures, the optimization of the process parameters becomes a challenging task. The success of a press hardening cycle results in a component with correct geometry (no distortions) and correct mechanical properties; both aspects are related to obtaining a successful quench by introducing a sufficiently hot blank in a well-adjusted die and keeping pressure for long enough to ensure that the component is cooled below the finish of martensitic transformation, typically below 200 °C [2].

In order to avoid any cooling of the part before forming, the blank must be transferred as quickly as possible from the furnace to the press; additional operations such as die opening and component extraction are also limited by the velocity of the available equipment. Therefore, the main variable that can be modified to impact cycle time is closed die time during quenching, which is commonly approximated by experience. Thus, cycle time reduction depends on the optimizations of the forming and quenching phases.

The use case to be studied is the hot stamping of a flat blank in a set of water-cooled flat dies. The problem modeling has been defined based on the STaMP pilot plant located in Eurecat Manresa [22] with the future aim of deploying and evaluating the experimental results in a real case. Figure 2 shows the test setup from the pilot plant used as a reference in this study. In the pilot, the die is mounted on a hydraulic press and cooling processes, where forming and metallurgical heat-treating take place during the stamping process. The furnace to heat the parts is located on the left of the image, and the hydraulic cooling die is on the right. The die is formed by steel DIN 1.2344, which is associated with AISI H13, and it is tempered at 48 ± 1. Two water channels of 10 mm in diameter are located at 20 mm depth from the surface with the objective of cooling the die. More details about the finite element analysis model and boundary conditions can be found in [21].

In this work, it is considered a 22MnB5 sheet with 1.7 mm thickness. No shape modification is considered in the hot stamping simulations since it is not a relevant difficulty in achieving the objectives of the study. The method presented can be extrapolated to a simulation environment of any shape by the execution of the training procedure similarly. This can be easily performed because the RL algorithm does not have any kind of initially integrated knowledge of the process.

The problem is presented as a simplified system consisting of a flat die quenching of flat blanks. This system foregoes plastic deformation and friction, greatly reducing complexity and turning the problem into essentially a heat transfer analysis. The batch size production is defined to be of 50 parts. A successful process part has been defined as a component extracted from the die at a temperature under 150 °C; this temperature, well below the finish of martensitic transformation, has been defined as an arbitrary threshold meant to generate a challenging optimization problem. Considering a constant initial blank temperature of 800 °C, this has been accomplished by monitoring several variables and acting on process control parameters. The cooling or resting time for the die is considered constant and equal to 10 s. Therefore, the optimization is made on the forming time defined in a range between 20 and 30 s, both included. This time can vary by 0.5 s per cycle, and the control model must decide whenever it is better to increase or decrease it in the next cycle. Twenty-one different initial forming times (from 20 to 30 s by increases of 0.5 s) condition the others since only modifications to the previous time are allowed. In order to cover all range of times within the parameters’ definition, the control training is performed considering random initial starting times, and evaluations are made on all initial times ensuring that all situations have been learned correctly. The restriction to 0.5 s variation is considered given that the pilot plant does not allow large changes from cycle to cycle. The total cycle time per part is the sum of the cooling and forming times, thus will be comprised between 30 and 40 s.

If the desired final part temperature of 150 °C is not reached, the part is not considered satisfactory, and the cycle will not be successful. Finally, the initial die temperature at the first cycle from the batch is fixed at 25 °C. Die temperature increases during cycles; however, to avoid die damage, it cannot go over 550 °C. The use case parameters definition is summarized in Table 1.

Due to the industrial limitations in the learning phase, the RL control system has been trained offline within a process simulator in order to demonstrate the RL capabilities. These limitations are related to (i) large real execution times, (ii) the large cost of failures of non-optimal policies in real production control, and (iii) the risk of safety industrial constraints violation. The RL model needs a large number of experiences to properly learn. A process simulator allows running many iterations or trials without jeopardizing real equipment or affecting real production planning. The simulation of press hardening has been performed by training surrogate models using transition data from ABAQUS simulations [20]. This software does not include specific solutions for press hardening; instead, it relies on user knowledge. However, it offers a good possibility of model adjustment and tweaking.

### 2.2. Dynamic Optimization Approaches

#### 2.2.1. Markov Decision Process

Optimal control is developed to find the most efficient ways for decision-making in dynamic and uncertain systems based on cost function definitions that represent the relevant concerns and desired final goals [23]. 

Sequential decision-making situations, where actions influence immediate rewards, future rewards, and subsequent system states, can be mathematically formalized as Markov Decision Processes (MDP). An MDP is a discrete-time representation of a process in which transitions between states are partly dependent on actions taken through the process and partly dependent on unforeseen environment dynamics. Manufacturing applications involve decision-making, parameter setting, and the control of processes where stochasticity is continuously changing the production scenario. These systems change over discrete or continuous times and require an understanding of the process dynamics to be able to make successful decisions. Moreover, future rewards can be discounted with the aim of giving more relative importance to closer situations than to the most distant ones. 

Formally, the MDP consists of a 5-tuple [24]: A set of states and actions $\left(S,\;A\right)$, a reward function R⊂ℝ, a state transition probability distribution p (from all states s∈S to all their successor $s{}^{\prime }\in S$), and a discounted factor $\gamma \;\in \;\left[0,\;1\right]$. The discounted factor γ allows for managing the trade-off between immediate and delayed rewards. With a discount factor γ=0, non-delayed rewards are considered, whereas, with a discount factor γ=1, all future rewards in time are valued within the same weight. γ=1 can only be considered in episodic environments [25]. A policy π is a function that maps states to actions based on expected rewards. The optimal policy is usually learned by approximating a value function that predicts a value to each state or state-action pair that summarizes future expected rewards.

A sequence of transitions, such as $\left\{s_{0},\;a_{0},\;r_{1},\;s_{1},\;a_{1},\ldots ,r_{49},\;s_{49},\;a_{49},r_{50},\;s_{50}\right\}$, defines a trajectory through the MDP. The objective of an MDP is to maximize the expected discounted cumulative reward in a trajectory by learning an optimal policy $\pi_{*}$. 

#### 2.2.2. Dynamic Programming: Value Iteration Method

In decision making, if the environment dynamics are completely known, that is, the transition probability distribution p; a policy π can be learned and improved by performing computations through the model until converging to an optimal policy $\pi_{*}$. Therefore, if state space and action space dimensions are tractable, the value and policy functions can be learned by the use of DP methods.

Classical optimal control solutions, such as DP, are considered offline and are computed through a prior and complete knowledge of the environment dynamics. DP refers to a collection of algorithms that can be used to find optimal policies given perfect models of the environment by the learning of value functions to organize and structure the search [24]. Although DP ideas can be applied to problems with continuous state and action spaces, exact solutions are possible only in special cases. These methods are not able to react to unexpected changes, handle uncertainties favorably, and make immediate real-time decisions in unknown or unplanned scenarios [26].

The V-table is a table that stores a value per each state of the MDP. The V-table with the approximate values is learned by iterative updates, and it is used to compute the (near) optimal policy π for decision-making by choosing the action that maximizes expected transition values.

More detail about the value iteration implementation is given in Appendix A. The computational cost of Algorithm A1 from Appendix A grows exponentially with the number of states, which makes it unfeasible to be applied to very large problems. Therefore, this methodology is known to suffer from the curse of dimensionality. 

#### 2.2.3. Reinforcement Learning: Q-Learning Method

Adaptive control allows making online real-time decisions without knowing the dynamics of the systems in more complex scenarios and in changing environments under uncertainties. RL is being increasingly used for dynamically adapting control strategies in unknown systems dynamics, providing optimal online solutions in real-time [27]. RL methods can be viewed as approximate dynamic programming, with less computation, higher dimensions, and without assuming a perfect model of the environment [28]. The pure RL appears when the environment is initially unknown, and no previous knowledge is given to the learning model. Therefore, the learning model has to interact with the environment to sequentially improve a policy by trial-and-error experience. The adaptative and immediacy essence of such methods offers great potential to be used as a decision support system or to directly manage, in an autonomous manner, decision processes.

RL provides smart, dynamic, and optimized process control supporting decision-making by automating goal-directed learning. Similar to DP, RL is learning what action to do on a given state; thus, it learns a function that maps states to actions with the aim of maximizing a numerical reward signal. However, in this case, environment dynamics are not necessarily known. 

In the learning process, the decision-maker is called the agent who interacts with the environment. These elements interact continually through an iterative process where the agent selects actions, and the environment responds to these actions with reward signals and presents new situations to the agent. This process is presented in Figure 3. The agent continuously improves a policy to maximize the expected future reward over time through its choice of actions. The ultimate goal of the agent is to learn a policy π that maps states to actions with the aim of generating the highest cumulative reward through the agent-environment interaction. The iterative learning process can be described as follows:

At each time-step t:The agent observes the current state of the environment $s_{t}$ and chooses an action $a_{t}$.This action causes a transition between states in the environment, which provides the new state$\;s_{t+1}$ and a reward signal $r_{t+1}$ to the agent.The agent updates a policy π based on the previous experience, denoted by the transition tuple $\left\{s_{t},\;a_{t},\;r_{t+1},\;s_{t+1}\right\}$.The current state $s_{t}$ is updated with the next one $s_{t+1}$.The process starts again until the number of iterations is completed. The process is reset whenever a terminal state is reached.

In a similar way to DP, the classical tabular Q-Learning control algorithm learns a state-action value function storing the expected values from each state-action pair received from experience in the Q-table. Recall that in the DP value iteration method, a value is learned per each state. In Q-Learning, a value is learned per each state-action pair.

During the training phase, new unknown states must be visited to enable a better and closed-to-reality generalization of the approximated value function. This represents a trade-off in the learning procedure when selecting actions to collect experiences during the agent-environment interaction. Therefore, the policy used to collect training experiences should be more exploratory than the learned behavior policy, also called the greedy policy. Exploitation consists of taking action following the Q-table values, which are assumed to be optimal with respect to the current knowledge. Exploration consists of selecting a random action, which can sometimes lead to poor decisions but also makes it possible to discover better ones if there exists any. The method used for tackling the exploration–exploitation dilemma in the data collection process is the epsilon greedy decay exploration strategy. At every time-step, random action is chosen with an epsilon probability. Otherwise, it is taken as the best-known action in the agent’s current state. The epsilon value starts at 1, and it decreases until a fixed minimum value allowing more exploration in the early stages and much more exploitation at the end of the training phase. 

The pseudo-code of the Q-Learning algorithm is illustrated in Algorithm A2 from Appendix A with more implementation details. The Q-Learning method also suffers from the curse of dimensionality as DP. Therefore, to manage and learn from more complex environments, Deep Q-Learning (DQN) algorithms [29] introduce Artificial Neural Networks (ANNs) for nonlinear value function approximations. In this context, the value function is approximated by ANNs, and it is used similarly to define the policy by selecting the action with the highest expected reward.

### 2.3. Problem Modelling

The MDP state must be carefully designed in order to contain all the required information to ensure that the control models can provide consistent decisions by identifying the necessities in each situation. The sequential decision-making problem deals with a stationary and deterministic environment. Since we have defined a finite batch size, it can be formalized as an undiscounted, episodic, and fully observable MDP. The episodic problem contains 51 time-steps *t* = {0, 1, …, 49, 50} which correspond to the remaining parts to produce, from 50 in *t* = 0 to 0 in *t* = 50. The overall production time of the 50 parts batch is considered the problem to be dynamically optimized, guarantying the temperature restrictions. 

#### 2.3.1. State Space

The MDP state must be correctly designed and contain all the required information to ensure that the control model can provide consistent decisions by identifying the necessities in each situation. With this objective, the state has been defined by a 3 parameters vector summarized in Table 2:The die temperature after the cooling phase: This temperature is the one at which the die is just before starting the forming and quenching phase. The lower the die temperature is, the more heat it will be able to absorb from the sheet. The range of die temperatures is between 25 °C and 550 °C. The lowest temperature corresponds to the initial one, and the highest is the maximum allowed for avoiding die damage.The current forming time setting: Since the action is based on increasing, decreasing, or maintaining a certain forming time, this forming time must be informed to make the decision. Thus, the previous forming time setting is an essential variable in the state. This variable can take values from 20 to 30 s with 0.5 s of granularity.Remaining parts from the batch: The aim is to produce 50 parts in the minim time possible without affecting part quality. Therefore, the remaining parts are a signal of the production state that tells how close the process is to the end. The range of values is between 50 parts in all initial states and 0 parts in all terminal states. In every episode step, this variable is reduced by one.

#### 2.3.2. Action Space

The action space defines the decisions to be taken in the environment. The decision-making is based on three discrete actions presented in Table 3. The forming time setpoint is bounded: 20 s≤forming time≤30 s. Thus, whenever it is not possible to increase the setpoint, action 1 will have no effect; and whenever it is not possible to decrease the setpoint, action 2 will have no effect. The description of the actions is described below, where $t_{f}^{c}$ refers to the current cycle forming time and $t_{f}^{c-1}$ to the forming time of the previous cycle:Action 0: Maintain the forming time from the previous cycle as shown in Equation (1).
(1)$$t_{f}^{c}=t_{f}^{c-1}$$

Action 1: Increase the forming time from previous cycle by 0.5 s as shown in Equation (2).


(2)
$$t_{f}^{c}=\min\;(30,\;t_{f}^{c-1}+0.5)$$


Action 2: Decrease the forming time from previous cycle by 0.5 s as shown in Equation (3).


(3)
$$t_{f}^{c}=\max\;(20,\;t_{f}^{c-1}-0.5)$$


The discrete action space with 3 actions presents a simpler problem than higher-dimensional discrete action spaces or a continuous one, either on the computation requirements or in the solutions space, which makes it suitable for initial developments, but it is probably too limited and restricted for an operative solution. However, the action space definition has been performed according to the pilot plant limitations. 

#### 2.3.3. Reward Function

The design of the reward function is one of the most delicate and complicated parts of the learning system, as it must guide the agent to the right path. It must express how good it is to take an action on a particular state with a unique numerical value. In this case, it is equivalent to a penalty function or a Key Performance Indicator (KPI) computed at each cycle of the process that needs to be maximized since it is expressed with a negative value. This cycle KPI is used for training the control models. Later on, a batch KPI based on the aggregation of the cycle in a production batch will be defined for policy evaluation.

Usually, control problems are based on multi-objective optimization; thus, more than one goal has to be taken into account to be accurately expressed in the reward signal. This implies balancing the relative importance of each concept. In the presented problem, the aim is to minimize the batch production time while ensuring parts quality and non-exceeding die temperature restrictions. The KPI presented analyses and penalizes the progress and operational improvements in terms of productivity and quality results. 

These goals are translated into penalties and aggregated in a unique reward signal received after each step (or cycle) and expressed in Equation (4).
(4)$$r\left(s,\;a\right)=-{\left({\mathrm{Cycle}}_{time}+100X_{p}+10^{6}X_{d}\right)}_{\left(s,a\right)}\;$$
for all s∈S and a∈A, and where $X_{p}$ is 1 if the final part temperature is above 150 °C, 0 otherwise, and $X_{d}$ is 1 if the die maximum temperature achieved is above 550 °C, 0 otherwise. Restrictions are translated to an approximate amount of pain, aggregated with the cycle time, and negated to be expressed as penalties since the objective is presented as a maximization problem. A non-accepted final part is penalized with a value of 100, which corresponds to approximately 3 cycles, and if the die exceeds the temperature threshold with the very high value of $10^{6}$ since it is a non-permitted situation. A feasible solution does not exceed any of the threshold temperatures defined. The reward components description is presented in Table 4.

#### 2.3.4. Control Agent

The control agent is the decision-making model that, given a state, immediately decides an action to achieve the goals that constitute the reward function. This decision-making process is illustrated in Figure 4. Two agents considering DP and RL have been developed. A brief summary of what has been explained extensively in previous sections is presented below:The DP agent is trained following Algorithm A1 from Appendix A. It learns the state value function in a tabular form by applying the dynamics from the environment to all states iteratively until convergence.The RL agent is trained following Algorithm A2 from Appendix A. It uses Q-Learning to learn the state-action value function in a tabular form by trial-and-error experience without knowing any environment dynamics. The stopping criteria for this learning process is a predefined parameter that indicates the number of episodes to experience.

#### 2.3.5. Environment Simulator

Real industrial environments are slow, present high costs, and have many risks related to safety conditions and failure. In this work, execution conditions and parameters are defined based on the capabilities of the pilot plant in Eurecat since future work will focus on a demonstrative phase in a real environment. Dynamic transitions are simulated with the determined setup within a software process simulator. Table 1 presents the experimental setup in which the formed part closing time is the decision variable. However, this software simulator is still too slow for the training of control models. Therefore, a data-driven approach has been taken to predict state process transitions.

These predictions are based on ML models. Three Extreme Gradient Boosting surrogate models have been trained to predict the needed system variables and act as the simulation environment of the industrial process. The prediction models for the continuous environment simulation with its inputs and outputs variables are presented in Figure 5 and explained below:Die maximum temperature model: Computes the prediction of the maximum temperature that the die reaches during the heat exchange process. This temperature is computed to check and avoid any damage to the die. Therefore, it is used as a penalization in the reward function.After-resting-die temperature model: Computes the prediction of the die temperature just before starting the heat exchange process. It is a state variable used for the decision-making of forming times.After-forming-part temperature model: Computes the prediction of the part temperature after the forming and quenching phase. This temperature must be below a defined target threshold to ensure the desired part quality.

The three data-driven models predict continuous variables. In this context, DP and Q-Learning cannot be applied since they are tabular methods. Therefore, the state space has been simplified by discretizing the outputs of the predictive models obtaining finite and manageable state-space dimensions.

## 3. Results

In this section, the experiment results are presented. DP and Q-Learning algorithms have been applied in the data-driven environment. The first algorithm learns by knowing all the dynamics; thus, it can obtain optimal solutions. The Q-Learning algorithm learns by experience, without any previous knowledge about the environment, which makes this learning process more challenging but more general. Both algorithms are compared in order to prove how Q-Learning, and in general RL algorithms, can obtain optimal solutions in this task. DP has memory and computation limitations since it explores all states. Recall that RL is a set of algorithms that can be applied to high-dimensional and continuous action and state spaces. Although the method used in this research is tabular, other RL methods with function approximation, such as DQN, should be explored. 

Hence, two experiments have been carried out, which are summarized in Table 5. Note that the number of iterations per experiment is presented in the table. The DP iterations depend on the epsilon threshold parameter defined in the setup, and each iteration refers to traversing the entire state space of the problem (see Algorithm A1 from Appendix A). On the other hand, the Q-Learning iterations parameter refers to episode executions (each episode has 50 steps), and it is defined in the setup before the training execution. Moreover, since there are 21 different initial start forming times, each new episode in training has a random initialization of this variable to allow exploration during training phases.

Considering that all transitions from the environment are already known, DP is straightforwardly applied by using Algorithm A1 from Appendix A with the parameters defined in Table 6.

Table 7 presents the setup values for the Q-Learning executions. The epsilon value is reduced linearly until it reaches the minimum value defined in the table. The decay value is computed in order to reach the minim value in the last episode.

The KPI for benchmarking used to measure the quality and improvements of the Q-Learning solution in managing the hot stamping cycle time is computed in terms of complete batch production. This indicator is expressed as a cost function, including the batch production time as the sum of all the cycle time from each part (performance cost) and the quality of the produced parts (quality cost) (see Figure 6).

There is a trade-off between these costs, and thus, the optimal solution balances the cycle time to maximize productivity with the resulting quality avoiding raw material loss. The cost indicates the capability and acceptance of the total production process. It is computed by the sum of the penalty for non-feasible parts produced. Directly derived from Equation (4), Equation (5) expresses the total batch cost calculation. Recall that the reward (or cycle penalty) is computed by the aggregation of the resting time and the forming time penalties, the penalty for non-feasible parts, and the penalty for exceeding the die permitted temperature.
(5)$$\mathrm{batch}\;\cos t=\underset{part}{\sum }{\left({\mathrm{Cycle}}_{time}+100X_{p}+10^{6}X_{d}\right)}_{part}\;$$

The Q-Learning agent, trained along 5 × $10^{5}$ episodes, has learned an optimal policy from all initial states. Figure 7 presents the penalty per batch evolution during training (from transforming Equation (5) to negative). At the start of the training, the batch cost evolution is highly oscillating due to exploration, but with a clear growing trend that shows how the agent is learning. It is not stable and goes up and down depending on the experiences of the agent, which is quite common in this type of learning. At the end of the training, this function converges and stabilizes since no more exploration is carried out, providing an optimal policy. The epsilon value through training episodes is also plotted in Figure 7. This variable value decreases during training and represents the degree of exploration in training episodes. Recall that the exploration technique used in these experiments is a linear epsilon greedy decay exploration strategy (shown in the secondary axis). 

DP and Q-Learning policies are benchmarked with the business-as-usual (BAU) policy in order to evaluate these approaches and their benefits. The BAU policy is defined by expert human knowledge that fixes a constant setpoint value for batch execution. The constant forming time value guarantees feasible parts. 

Table 8 presents the batch cost production for all the experiments, hence considering all initial forming times for each of the policies. A non-feasible solution is marked in red, and the best solutions within each policy are marked in green. DP and Q-Learning costs obtained are the same in all experiments, which validates the Q-Learning solution. Moreover, this comparison demonstrates the Q-Learning policy optimality. The savings with respect to the BAU policy are presented in the last column. This policy is evaluated for all the possible forming times to find the minimum fixed time value that produces good parts.

For an initial forming time of 20, 20.5, and 21 s, there is no feasible solution since some parts are produced with a temperature higher than the maximum allowed defined of 150 °C. The rewards in this execution are being penalized by the cost of the part cooling temperature not being achieved. The other initial forming times can guarantee parts quality with dynamic forming time policies. With the BAU policy, feasible solutions are guaranteed for a forming time equal to or higher than 27 s. Moreover, all computed solutions preserve the die temperature under the permitted threshold of 550 °C. Recall that at the beginning of the batch production, the die temperature is considered to be 25 °C. With this die temperature, the heat exchange reduces the part temperature very fast in the forming and quenching phase. However, the die temperature increases every cycle until it stabilizes, as it is shown in Figure 8. The RL (Q-Learning) policy is executed for a batch in the best scenario of 23 s of initial forming time, and reward, part temperature, and die temperature variables are plotted. Note that the left axis expresses times in seconds for the forming time variable and the right axis temperatures in °C for all the temperature variables. The stabilization phase in the figure is due to the assumption of constant variables: the initial temperature of a part after heating and the die resting time. In a more realistic setting, these parameters would have some variability, and the stabilization stage would not totally occur. In this context, an RL approach would increase time savings in the process. Similar to the die temperature, the part temperature increases each cycle until the stabilization phase occurs. Within this setting, time varies during the first 10 cycles, then stabilizes, and finally decreases during the last three cycles since higher temperatures can be assumed when there are no more parts to be produced. The forming time stabilizes in 27 s.

The difference between the best BAU cost solution and the best DP and Q-Learning cost solution is 24.5 s per batch. Figure 9 presents the execution of these policies for production with the comparison of forming times settings and the temperature of the final parts. Since DP and Q-Learning provide the same solution (both optimal), they overlap in the figures. Note that in Figure 9b, in DP and Q-Learning solutions, parts temperature increases during the last three cycles until the 150 °C threshold is reached. During non-stable temperature phases, savings are higher than 5% of the total cycle time. 

## 4. Discussion

This work is focused on optimizing the time used in the forming and quenching production step for hot stamping processes. This is performed by minimizing a cost-performance indicator defined based on balancing the productivity with the resulting quality avoiding raw material loss. The cost is used to measure production losses, detect process improvements, help in systematically analyzing the process, and benchmark the different policies.

The setup parameters have been designed with domain experts from the pilot plant in order to consider significant challenging scenarios that can be tested in real processes. Two approaches have been taken in a data-driven environment simulator. The first approach considered, a dynamic programming algorithm, gives an optimal solution. This approach is not scalable to high-dimensional problems. Therefore, new techniques need to be considered since real processes cannot be simplified in a finite and manageable number of state transitions. Reinforcement Learning is introduced in order to provide more scalable solutions, and Q-Learning is evaluated against DP achieving equivalent performance. Figure 10 summarizes the conceptual idea behind this research.

Online learning directly from the real systems cannot be considered at this stage since many non-desired situations are tested and would have bad consequences in real environments. Moreover, real processes take a too long-time execution. The software ABAQUS simulates the process, and it is used to generate experimental data to train ML models from the collected transitions. Working with the developed data-driven surrogate models as an environment for RL agents has many advantages that make it very interesting, such as a fast simulation cycle computation. The data-driven models predict continuous variables. In this context, a DP approach cannot be applied. The output of these models has been discretized to make the DP approach feasible in this environment. Q-Learning has a similar problem since, although it does not need to know the environment dynamics, it is a tabular method that stores a value per each state-action pair. Thus, both spaces need to be finite. For this reason and to allow comparison, the same discretization is considered for the Q-Learning training. The DP and Q-Learning models’ evaluation has been made by considering all initial states and comparing the solutions from both algorithms with the BAU policy. Similar performance results are obtained with both methods outperforming the BAU behavior. The BAU policy has been defined with the best-fixed value setting presenting the most optimistic time selection. It has been proven how RL control can achieve a 5% of cycle time reduction in non-stable phases. 

An RL algorithm with a function approximation to predict state-action values, such as a Q-Learning extension, does not need any data discretization. It allows to easily address real operation settings and to better generalize. When considering non-constant parameters, stable stages would not totally occur, and more dynamic times updates would provide better results. The guided cycle time control in real production lines can significantly reduce the operational costs and energy consumption and achieve challenging goals in the parts produced by a dynamic real-time time setting, avoiding material losses. Thus, when more complex and variable a process becomes, the more gains can be obtained by applying RL control. 

Despite all the advantages, some challenges have been carried out in this research. The challenges from these implementations are summarized in Table 9 and explained below: The curse of dimensionality [30] is overcome by considering a discrete action space and the simplification of the state space by the discretization of temperatures.The reward function definition is performed by different evaluations of functions to find the best guide for the models’ training. New goals would lead to a different reward function definition.The surrogate model’s error accumulation in prediction upon predictions for the models-based environment is reduced by evaluating different algorithms and training new versions able to improve generalization. It is important to recall that a data-driven algorithm acts as a function approximation of the dynamics embedded in the simulated process data.

The results provided in this paper represent a proof of concept for more complex and less constrained production systems.

## 5. Conclusions

We have shown how hot stamping processes performance can be improved by a dynamic time setting configuration while keeping outputs quality and avoiding raw material loss. A DP approach is used to demonstrate the optimality of the RL implementation, which is a more extendable technique. The benefits are evaluated against the BAU industrial behavior. The RL model outperforms the industry-standard control in a cycle time setting. Decisions are made based on the state and temperature parameters of the production process. The autonomous control is guided by a cycle KPI defined to manage the trade-off between productivity and quality results. A time reduction of 5% is achieved in the non-stable stages of the cycle. More variability in the process as real environments have will lead to an increase in the benefits of a dynamic approach. 

Further research will be performed in consideration of more dynamic variables in the forming process, such as the initial blank temperature for the stamping phase, to extend this conclusion to more complex systems and demonstrate the RL advantages in a more realistic case study with higher optimization needs. This problem extension cannot be solved with tabular algorithms; thus, more complex RL methods with function approximations will be considered, such as the value-based Deep Q-Learning, or actor-critic methods such as Soft-Actor Critic. The final goal of this research is to apply the control models to manage the pilot plant from Eurecat.

## Figures and Tables

**Figure 1 materials-15-04825-f001:**
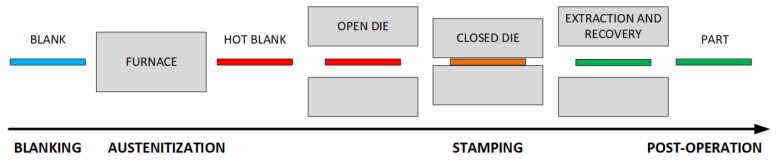
Hot stamping direct method process phases.

**Figure 2 materials-15-04825-f002:**
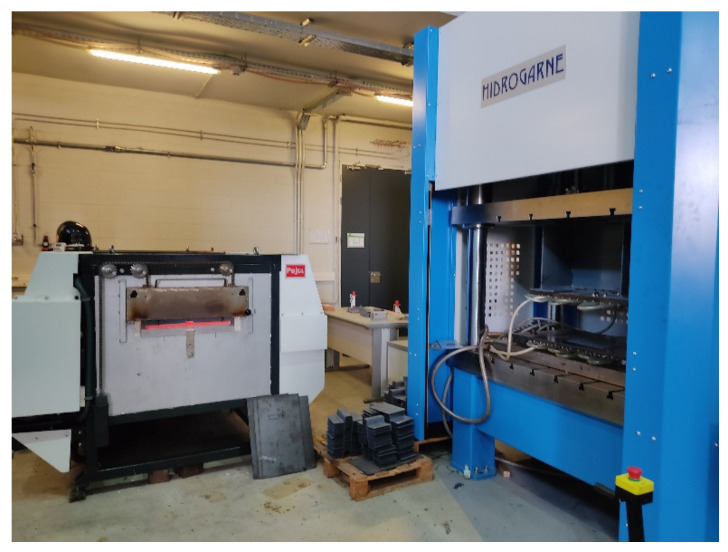
The pilot plant at Eurecat Manresa from which is defined the experiment considered in simulation and control.

**Figure 3 materials-15-04825-f003:**
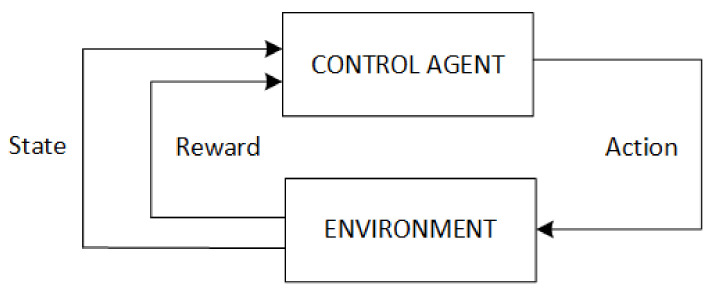
Reinforcement Learning interaction process scheme.

**Figure 4 materials-15-04825-f004:**
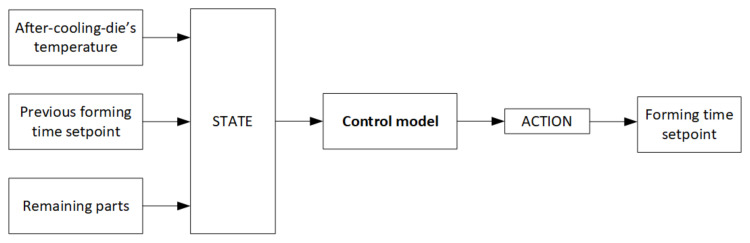
Workflow of the control model.

**Figure 5 materials-15-04825-f005:**
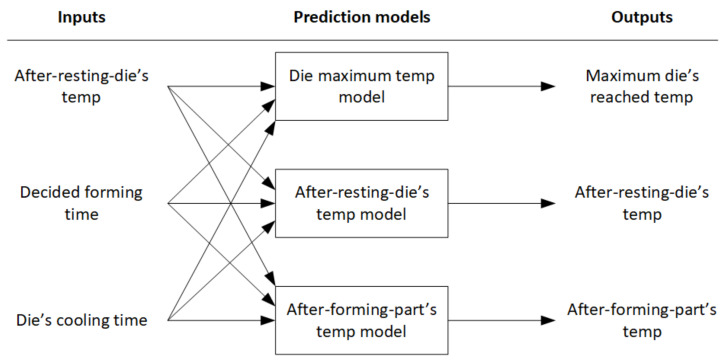
Data-driven prediction models for the continuous environment simulation.

**Figure 6 materials-15-04825-f006:**
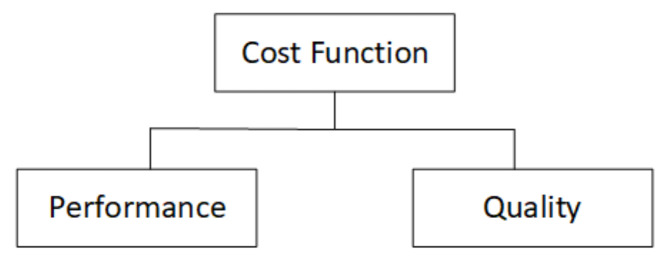
Cost function definition to evaluate the hot stamping process.

**Figure 7 materials-15-04825-f007:**
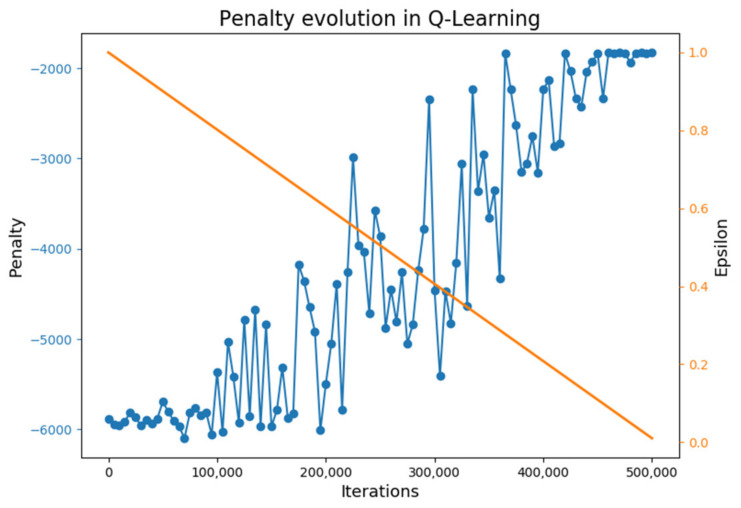
Q-Learning agent’s policy improvement during training.

**Figure 8 materials-15-04825-f008:**
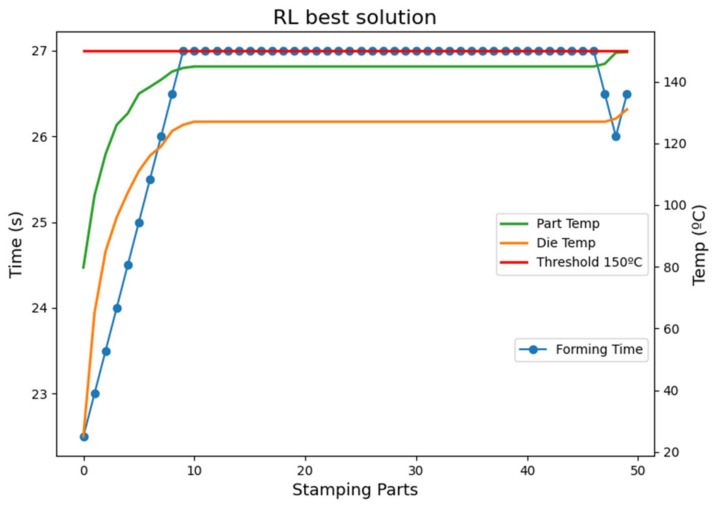
Batch execution with RL (Q-Learning) best solution.

**Figure 9 materials-15-04825-f009:**
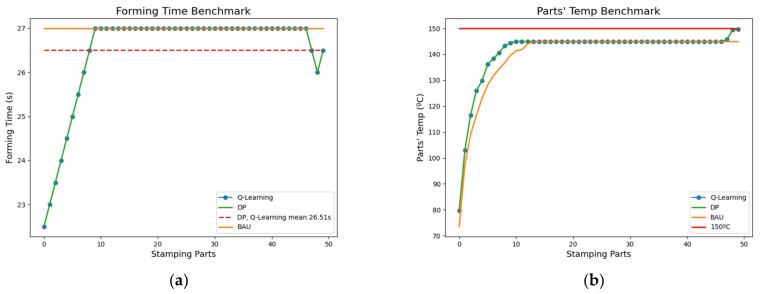
Best BAU, DP, and Q-Learning solutions benchmark for a batch execution; (**a**) Forming time setting benchmark, (**b**) Final parts temperature benchmark.

**Figure 10 materials-15-04825-f010:**
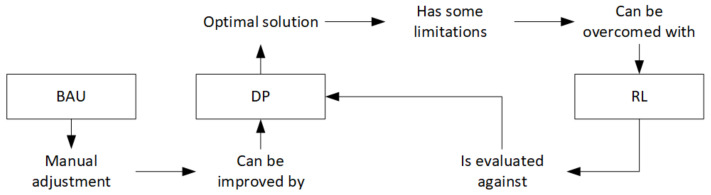
Models of cycle time adjustment and its connections considered in this work.

**Table 1 materials-15-04825-t001:** Parameter range setting in the hot stamping process case study.

Parameters	Use Case Values Definition
Batch size	50 parts
Die initial temperature at the beginning of the batch	25 °C
Die temperature allowed	≤550 °C
Blank initial constant temperature	800 °C
Formed part closing time	20–30 s
Constant recovery time	10 s
Formed part cooling temperature	≤150 °C
Cycle time	30–40 s

**Table 2 materials-15-04825-t002:** Description of the MDP state space.

State Variables	Range of Acceptable Values
Die temperature	25–550 °C
Current forming time setting	20–30 s
Remaining parts from batch	0–50 parts

**Table 3 materials-15-04825-t003:** Description of the MDP action space.

Discrete Actions	Forming Time Variation at Each Step
0	0 s
1	+0.5 s
2	−0.5 s

**Table 4 materials-15-04825-t004:** Description of the MDP reward function.

Cost Function Components	Range of Penalties Values	Condition
Cycle time	30–40	Always
Part cooling temperature not achieved	100	$\mathrm{If}\;{\;\mathrm{temp}}_{\mathrm{part}}$ > 150 °C
Maximum die temperature exceeded	$10^{6}$	$\mathrm{If}\;{\;\mathrm{temp}}_{\mathrm{die}}$ > 550 °C

**Table 5 materials-15-04825-t005:** Description of the experiments presented.

Experiment	Environment	Algorithm	Iterations
Experiment 1	Data-driven models-based	DP	50
Experiment 2	Data-driven models-based	Q-Learning	$5\;\times \;10^{5}$

**Table 6 materials-15-04825-t006:** Dynamic Programming parameters setup values.

DP Parameters	Setup Values
Gamma	1
Epsilon	$10^{-3}$

**Table 7 materials-15-04825-t007:** Q-Learning parameters setup values.

Q-Learning Parameters	Setup Values
Episodes	$5\;\times \;10^{5}$
Gamma	1
Alpha	0.1
Initial epsilon value	1
Epsilon decay	Linear
Minimum epsilon value	0.01

**Table 8 materials-15-04825-t008:** Batch penalty production for all the experiments with the BAU, DP, and Q-Learning policies. A non-feasible solution is marked in red, and the best solutions within each policy are marked in green.

Initial Forming Time	BAU Penalty	DP Penalty	Q−Learning Penalty	Savings
20	**−6000**	**−3240**	**−3240**	2760
20.5	**−5925**	**−2841**	**−2841**	3084
21	**−5950**	**−1927**	**−1927**	4023
21.5	**−5975**	−1832.5	−1832.5	4142.5
22	**−5900**	**−1825.5**	**−1825.5**	4074.5
22.5	**−5925**	**−1825.5**	**−1825.5**	4099.5
23	**−5850**	**−1825.5**	**−1825.5**	4024.5
23.5	**−5775**	−1826	−1826	3949
24	**−5800**	−1826.5	−1826.5	3973.5
24.5	**−5825**	−1827.5	−1827.5	3997.5
25	**−5750**	−1828.5	−1828.5	3921.5
25.5	**−5575**	−1830	−1830	3745
26	**−5600**	−1831.5	−1831.5	3768.5
26.5	**−5525**	−1833	−1833	3692
27	**−1850**	−1835	−1835	15
27.5	−1875	−1837	−1837	38
28	−1900	−1839.5	−1839.5	60.5
28.5	−1925	−1842	−1842	83
29	−1950	−1843.5	−1843.5	106.5
29.5	−1975	−1847	−1847	128
30	−2000	−1851	−1851	149

**Table 9 materials-15-04825-t009:** Challenges in DP and Q-Learning implementations.

Challenge	Solution
Curse of dimensionality	Discrete action and state spaces definition
Reward function definition	Evaluation of different reward functions
Surrogate models error	Evaluation of different algorithms

## Data Availability

The data presented in this study are available upon reasonable request from the corresponding author.

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
