# Peer review of "A Reinforcement Learning Control in Hot Stamping for Cycle Time Optimization"

_materials, 2022, doi:10.3390/ma15144825_

Round 1

Reviewer 1 Report

The paper describes a methodology for improving Hot Stamping process based on data analysis. It lacks important details in the methods section. It is also not clear what the novelty of the work is. The scientific level is also considered as low since the work looks more like a practical application of reinforcement learning (in a very general sense).

The following are the points that the reviewer thinks must be improved to be considered for publication:

- The text is overcrowded with unnecessary explanations or non-scientific opinions. For instance the first 3 lines of the abstract (opinionated). The first paragraph of introduction, (no references) etc. Please remove these.

- There not enough references that would help put the novelty of the work in perspective and also to justify the methods that are chosen. The used methods and their capabilities are also not explained.

- The novelty of the paper is unclear. It looks like applying reinforcement learning (a very general term) to hot-stamping with a limited number of trials.  

- Section 2.2 (and especially 2.2.3. where RL is explained) is the only part (with the Appendix) where the most important scientific content is to be found. And it only gives a very general outline of the method. Therefore it is difficult to judge if the methods are applied correctly or not.

- The training simulations are not given and therefore the suitability/correctness/accuracy of the training is impossible to judge. The sentence where it is claimed 'LS Dyna in ABAQUS is used' is meaningless since they are two different software applications both to do simulation of thermomechanical forming processes. 

- The part about the environment simulator is unclear. There are also newly introduced ML models, talk about Digital Twins suddenly appearing without clearly explaining the methods and the relevance to the research.

Overall there is no clear presentation of the method including the training simulations. Only abstract equations are presented with accompanying text which does not help explain the methods and therefore the methods remain obscure. 

Reviewer 2 Report

The article deals with a very relevant topic for the moment. It has high originality and employs modern technologies and high technological potential. The structure of the article is excellent, well-founded and solid. The article navigates through the areas of mechanical forming, process control and computing.

Reviewer 3 Report

This paper shows the results of a study that applied Reinforcement Learning to hot stamping process data to reduce the cycle time of hot stamping. This paper can be considered as a meaningful study in that it shows that the optimal conditions for the hot stamping process can be found through reinforcement learning such as Q-Learning.

However, the following issues need to be addressed.

1. The detailed description of hot stamping parts and hot stamping process needs to be further supplemented. For example, there are no drawings or photographs showing the shape of the part. Since the temperature and cycle time settings during the process vary depending on the part, information about the part shape is very important.

2. In addition, it is necessary to supplement the details of finite element analysis model and boundary conditions for hot stamping analysis using ABAQUS.

3. In order to check whether the optimal conditions derived from this paper are actually optimal, it is necessary to present corresponding experimental data. If you can confirm the reliability of the research results by means other than the experimental data, you may present them.

Round 2

Reviewer 3 Report

Figure 1 seems to have been modified to Figure 2. However, in the manuscript, Figure 1 and Figure 2 are shown together. It is presumed that the problem occurred when the author used track change and saved it as a pdf file. Please check the figures and figure numbers and correct them.
